# UNSUPERVISED DOMAIN ADAPTATION VIA FEATURE ALIGNMENT AND REDUNDANCY REDUCTION

## ABSTRACT

Unsupervised domain adaptation (UDA) addresses the challenge of transferring knowledge from a labeled source domain to an unlabeled target domain for the same task under data distribution shifts. Current approaches rely on strong hypotheses about the type of domain shift, or task to perform. We propose FARR, a novel UDA method with 3 main contributions: (i) a new feature-alignment strategy based on redundancy reduction, that is task-adaptable and agnostic to the type of domain shift; (ii) a theoretical proof that our formulation effectively aligns source and target features; (iii) a comprehensive empirical evaluation across classification and segmentation tasks, using seven public and two private datasets covering diverse domain shifts. Our results show that FARR consistently outperforms existing feature-alignment methods, while remaining competitive with state-of-the-art UDA approaches across tasks and datasets. Our code is available at [1].

## 1 INTRODUCTION

Modern deep learning models achieve impressive performance across a wide range of tasks, yet their success is built on the availability of large-scale labeled datasets for supervised training. However, when the deployment dataset follows a different distribution from the training data, this domain shift hampers successful generalization. Unfortunately collecting a large annotated dataset in the target domain is unfeasible in many real-world scenarios, due to the time, cost, and expertise required. Domain adaptation, a subset of transfer learning, adresses this challenge in the case where source and target tasks agree. Here, we focus on Unsupervised Domain Adaptation (UDA), where annotations are only available in the source domain. Our goal is thus to exploit the annotated source data to learn a model that performs well on the unlabeled target domain.

UDA is a well-studied and actively evolving research area, where methods can be characterized as feature-alignment, image-alignment, or self-training-based. Feature-alignment methods such as Ganin et al. (2016), Tzeng et al. (2017), Javanmardi et al. (2018) learn a feature space and align the distributions of source and target data in that space. These methods are usually *restricted* to a specific task, as in Zhang et al. (2019). Image-alignment methods such as Chen et al. (2019), Hoffman et al. (2018) aims to directly align the data distributions in the pixel space. These methods can perform very well, but they are usually *specific* to a particular kind of domain shift. For instance, Hoffman et al. (2018) assumes a similar field of view between source and target images. Both feature-alignment and image-alignment methods can be *complex* to train and *unstable* as they usually require one or several GANs Fu et al. (2024), Kamnitsas et al. (2017), Jiang et al. (2021), Liu & Tuzel (2016). Chen et al. (2019). However, image-alignment methods tend to be heavier Jiang et al. (2021). Other approaches aim to generate pseudo-labels in target domain to train a model under the supervision of these pseudo-labels Zhou et al. (2021). These methods implicitly make the hypothesis that the domain shift is limited, as they require an initialization of the network that must produce pseudo-labels of sufficient quality Lee et al. (2022).

To the best of our knowledge, there is no method that is task-adaptable, agnostic to the type of domain shift, that benefits from a rigorous theoretical background, and that achieves state-of-the-art performance.

---

[1] https://anonymous.4open.science/r/FARR-06B3

Segmentation task with U-Net                Classification task with ResNet

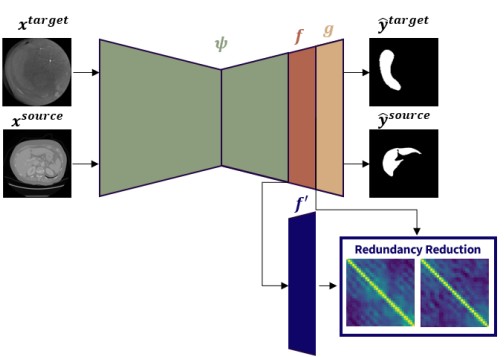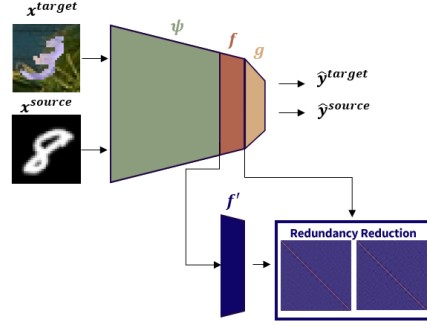

Figure 1: Our task-agnostic UDA method FARR. We decompose any architecture into a feature extractor $\psi$, a representation head $f$ and its adversary $f'$, and a task-specific prediction head $g$. The features $z = \psi(x)$ are encouraged to become invariant to the domain shift between the source and target data by introducing a redundancy reduction adversarial strategy. After training, $\psi$ extracts domain-invariant features and $f'$ is discarded at inference.

**Contributions.** Our contributions are: (i) a new feature-alignment strategy based on redundancy reduction, that is task-adaptable and agnostic to the type of domain shift; (ii) a theoretical proof that our formulation effectively aligns source and target features; (iii) a comprehensive empirical evaluation across classification and segmentation tasks, using seven public and two private datasets covering diverse domain shifts. Our results show that FARR consistently outperforms existing feature-alignment methods, while remaining competitive with state-of-the-art UDA approaches across tasks and datasets.

## 2 RELATED WORK

Existing UDA methods can be grouped into feature-alignment, image-alignment and self-training. Feature-alignment methods learn a domain-invariant embedding space aligning source and target features. Image-alignment methods operate at the pixel level, translating target images into the source style to reduce domain shift. Self-training methods instead generate pseudo-labels in the target domain, enabling supervised training with these labels.

**Feature-alignment.** Early work minimizes statistical distances such as MMD Long et al. (2015) or CORAL Sun et al. (2016), using hand-crafted losses. More recent approaches rely on adversarial training: DANN Ganin et al. (2016) introduces a domain discriminator to enforce invariance, later extended to segmentation tasks with U-Net backbones Kamnitsas et al. (2017); Javanmardi et al. (2018); Panfilov et al. (2019); Fu et al. (2024). ADDA Tzeng et al. (2017) improves stability with separate source and target encoders. MDD Zhang et al. (2019) proposes a margin-based discrepancy measure with theoretical guarantees, later adapted to medical segmentation Munk et al. (2024). Beyond adversarial learning, contrastive methods have been explored, combining distributional alignment with contrastive objectives Thota et al. (2021); Avena et al. (2025), or applying slice-based contrastive pairs for OCT segmentation Gomariz et al. (2022).

**Image-alignment.** These methods translate images across domains to reduce pixel-level discrepancies. CoGAN Liu & Tuzel (2016) and PixelDA Bousmalis et al. (2017) use GANs to generate target-like images from source data, sometimes with feature-level adaptation. To better preserve semantic consistency during translation, cycle-consistency is often employed Hoffman et al. (2018); Chen et al. (2019; 2020); Cai et al. (2019); Huo et al. (2019); Jiang et al. (2021). Extensions such as DRANet Lee et al. (2021) further disentangle structure and style to improve transfer. While such methods can produce strong results, they usually require stronger assumptions on the type of domain shift, lack theoretical support, and involve complex architectures with multiple adversarial modules.

**Self-training.** These methods exploit pseudo-labels generated on target data to iteratively adapt the model Zhou et al. (2021); Zhang et al. (2021); Chen et al. (2023); Liu et al. (2024). Performance depends heavily on pseudo-label quality, often controlled by confidence thresholds. Recent improvements integrate contrastive learning and prototypes Lee et al. (2022), clustering-based consistency losses Yue et al. (2023), or additional weak labels Das et al. (2023). While robust in moderate shifts, they may fail when initial pseudo-labels are highly noisy.

## 3 METHOD

Driven by the previous discussion, we aim to design a method that is generic and task-adaptable, and where we make no prior assumption on the type of domain shift. Furthermore, it should be flexible, namely no constraints about the architecture, and lightweight enough to train with minimal computational resources. Eventually, to enhance feature alignment, and provide a theoretical framework to ensure domain invariance, we will take advantage of the self-supervised learning loss of Barlow Twins Zbontar et al. (2021).

We consider a source domain $\mathcal{D}_S = (\mathcal{X}^S, \mathcal{Y}, p^S(x, y))$ and a target domain $\mathcal{D}_T = (\mathcal{X}^T, \mathcal{Y}, p^T(x, y))$ with some domain shift captured by differing distributions $p^S$ and $p^T$. We are given labeled samples $(x_i^S, y_i^S)_{i=1}^{N_S} \sim p^S(x, y)$ in the source domain, and unlabeled samples $(x_i^T)_{i=1}^{N_T} \sim p^T(x)$ in the target domain. The goal is to learn a map $h : \mathcal{X}^S \cup \mathcal{X}^T \to \mathcal{Y}$ that produces a consistent labeling $h(x)$ for both source and target domain samples $x \in \mathcal{X}^S \cup \mathcal{X}^T$. We proceed via a feature-alignment strategy, whereby (1) learned representations of source and target domain samples are driven to the same marginal distribution at convergence, and (2) the map $h$ is made informative for the labeling task through explicit label supervision in the source domain. Next, we describe an adversarial strategy, that we name FARR (Feature Alignment and Redundancy Reduction), that leads to these two properties both theoretically and empirically. To derive the proposed approach, we split the predictor $h(x) \triangleq g(f(\psi(x)))$ into a feature encoder $\psi(\cdot)$, a representation head $f(\cdot)$ and a task-specific prediction head $g(\cdot)$. Example architectures for $\psi, f, g$ are provided in Figure 1. An ablation study on splitting a ResNet-18 for classification task can be found in Appendix A.9.

**Source domain label supervision.** The representation head $f(\cdot)$ and the prediction head $g(\cdot)$ are trained to minimize the labeling task in the source domain via a task-specific loss $\mathcal{L}_{\text{task}}(h(x^S), y^S)$.

**Adversarial feature alignment.** The features $z \triangleq \psi(x)$ are encouraged to become invariant to the domain shift between the source and target domains, by introducing an adversarial strategy between $\psi(\cdot)$ and an adversary $f'(\cdot)$, where $f'$ shares the same architecture as $f$. The adversary $f'(\cdot)$ is trained to produce representations as dissimilar as possible from $f(\cdot)$ when using target samples $x^T$, and at the same time it should result in representations as similar as possible to $f(\cdot)$ when using source samples $x^S$. This is achieved (equation 1) via a separation loss $\mathcal{L}_{\text{sep}}$ and an alignment loss $\mathcal{L}_{\text{align}}$, respectively, described in Section 3.1. On the other hand, the feature extractor $\psi(\cdot)$ is trained (equation 2) to output features $z \triangleq \psi(x)$ that (1) are relevant for the labeling task; and (2) that make the representations of the adversary $f'(z)$ and of the representation head $f(z)$ undistinguishable both in source and target domains:

$$\arg\min_{f'} \mathcal{L}_{\text{align}}\left(f(z^S), f'(z^S)\right) + \gamma \mathcal{L}_{\text{sep}}\left(f(z^T), f'(z^T)\right) \tag{1}$$

$$\arg\min_{\psi} \mathcal{L}_{\text{task}}(h(x^S), y^S) + \alpha \mathcal{L}_{\text{align}}\left(f(z^S), f'(z^S)\right) + \gamma \mathcal{L}_{\text{align}}\left(f(z^T), f'(z^T)\right) \tag{2}$$

where $(\alpha, \gamma) \in \mathbb{R}^+ \times \mathbb{R}^+$ are user-defined hyperparameters.

**Algorithm.** In practice, we alternate between single optimization steps of $\mathcal{L}_{\text{task}}$, equation 1 and equation 2 using a simple stochastic gradient descent on source and target domain batches (see Algorithm 1). We will show in Section 3.3 that, at convergence, the source and target domains are aligned in feature space *i.e.*, the marginal distributions of $z^S$ and $z^T$ are identical.

## 3.1 Redundancy-reduction based representation alignment

The losses $\mathcal{L}_{\text{align}}\left(f(z), f'(z)\right)$ and $\mathcal{L}_{\text{sep}}\left(f(z), f'(z)\right)$ should encourage alignment and separation of the latent representations $f(z), f'(z)$, respectively. To this end, we revisit Barlow Twins' redundancy reduction mechanism Zbontar et al. (2021) and encourage the correlation matrix of representations $f(z), f'(z)$ to have specific structure conducive of these properties. Consider the representations $f(\mathbf{z}) \in \mathbb{R}^{B \times D \times H \times W}$ obtained from a batch, with $H := W := 1$ for classification and arbitrary $H, W$ for segmentation[2]. Let $\phi_f(\mathbf{z}) \in \mathbb{R}^{D \times N}$ the reshaped, flattened, centered and $L_2$-normalized (along $B, H, W$) version of $f(\mathbf{z})$, and similarly $\phi_{f'}(\mathbf{z})$ for $f'(\mathbf{z})$. Following Zbontar et al. (2021), we note $\mathcal{C}[i, j] \triangleq \langle \phi_f(\mathbf{z})_i, \phi_{f'}(\mathbf{z})_j \rangle$ the cross-correlation of features $i, j \in \{1, \cdots, D\}$ for representations $f(\mathbf{z}), f'(\mathbf{z})$.

**Alignment loss.** We define $\mathcal{L}_{\text{align}}\left(f(z), f'(z)\right)$ in equation 3 to encourage perfect correlation between the representation head $f(\cdot)$ and the adversary $f'(\cdot)$ along homologous feature dimensions, and to discourage redundancy across different feature dimensions:

$$\mathcal{L}_{\text{align}}\left(f(z), f'(z)\right) \triangleq \sum_{i=1}^{D}(1 - \mathcal{C}[i, i])^2 + \frac{1}{D}\sum_{i \neq j}\mathcal{C}[i, j]^2 \qquad (3)$$

**Separation loss.** On the contrary, we define $\mathcal{L}_{\text{sep}}\left(f(z), f'(z)\right)$ in equation 4 to encourage perfect decorrelation between the representation head $f(\cdot)$ and the adversary $f'(\cdot)$ across all feature dimensions:

$$\mathcal{L}_{\text{sep}}\left(f(z), f'(z)\right) \triangleq \sum_{i=1}^{D}\mathcal{C}[i, i]^2 + \frac{1}{D}\sum_{i \neq j}\mathcal{C}[i, j]^2 \qquad (4)$$

**Remark.** Note that $\mathcal{L}_{\text{align}}$ and $\mathcal{L}_{\text{sep}}$ are task-agnostic: they encourage representations to match or differ without direct reference to the task of interest *i.e.*, to the prediction head $g(\cdot)$. This is in contrast to previous works, such as MDD Zhang et al. (2019), where specific losses, such as cross-entropy for classification, were used. Furthermore, by using the proposed redundancy-reduction losses, we don't need an adversary task-specific prediction head $g'(\cdot)$, as in MDD, since we move the feature alignment from the space of $g$ to the one of $f$.

## 3.2 Learning algorithm

Using all the notations introduced above, we provide a description of the unsupervised domain adaptation algorithm.

---

**Algorithm 1** Proposed Unsupervised Domain Adaptation Algorithm

---

**Require:** Define hyper-parameters $\alpha, \gamma$, learning rates, batch size B, and number epochs $K$
    Pre-train $\psi$, $f$ and $g$ by minimizing $\mathcal{L}_{\text{task}}$ on source domain $S$ for $K$ epochs
    Initialize $f'$ with the same weights as $f$ pre-trained on source domain $S$
    **for** $k \leftarrow 1$ to $K$ **do**
        Sample batch $\{x_i^S, y_i^S\}_{i=1}^{B}$ from source domain and batch $\{x_j^T\}_{j=1}^{B}$ from target domain
        Update $f$ and $g$ by minimizing $\mathcal{L}_{\text{task}}$
        Update $f'$ by minimizing equation 1
        Update $\psi$ by minimizing equation 2
    **end for**

---

Note that $f'$ is only used during the unsupervised domain adaptation phase and discarded after training.

---

[2]in this article, we focus on classification and segmentation but the proposed method is generic and could be applied to any supervised task, such as regression or object detection.

## 3.3 THEORETICAL GUARANTEES

We now study the theoretical properties of the proposed method. In particular, we obtain theoretical guarantees that, at the optimum, the feature extractor $\psi(\cdot)$ realizes the alignment of the marginal distributions of features in the source and target domains, under Assumption 1.

Consider a minibatch $\mathbf{x} \in (\mathcal{X}^S \cup \mathcal{X}^T)$, and the marginal distributions $p^S(\mathbf{x}), p^T(\mathbf{x})$ according to the source and target domain distributions. Denote $p(\mathbf{z}) \triangleq \psi_\sharp p^S(\mathbf{z})$ the pushforward of $p^S$ through $\psi$, and similarly $q(\mathbf{z}) \triangleq \psi_\sharp p^T(\mathbf{z})$ the pushforward of $p^T$ through $\psi$.

We start by rewriting equation 1 with the losses proposed in Section 3.1, obtaining:

$$
\underset{f'}{\arg\min} \, \mathbb{E}_{\mathbf{z} \sim p} \left[ \sum_{i=1}^{D} (1 - \langle \phi_f(\mathbf{z})_i, \phi_{f'}(\mathbf{z})_i \rangle)^2 + \frac{1}{D} \sum_{i=1}^{D} \sum_{j \neq i} \langle \phi_f(\mathbf{z})_i, \phi_{f'}(\mathbf{z})_j \rangle^2 \right]
$$
$$
+ \gamma \mathbb{E}_{\mathbf{z} \sim q} \left[ \sum_{i=1}^{D} \langle \phi_f(\mathbf{z})_i, \phi_{f'}(\mathbf{z})_i \rangle^2 + \frac{1}{D} \sum_{i=1}^{D} \sum_{j \neq i} \langle \phi_f(\mathbf{z})_i, \phi_{f'}(\mathbf{z})_j \rangle^2 \right]
\tag{5}
$$

Then, we reformulate the optimization problem by substituting $w(\mathbf{z})_{i,i} = \langle \phi_f(\mathbf{z})_i, \phi_{f'}(\mathbf{z})_i \rangle$ and minimizing the resulting relaxed variational problem with respect to $w(\mathbf{z})$:

$$
\underset{w(\mathbf{z})}{\arg\min} \int_{\mathbf{z}} p(\mathbf{z}) (\sum_{i=1}^{D} (1 - w(\mathbf{z})_{i,i})^2 + \frac{1}{D} \sum_{j \neq i} w(\mathbf{z})_{i,j}^2) + \gamma q(\mathbf{z}) (\sum_{i=1}^{D} w(\mathbf{z})_{i,i}^2 + \frac{1}{D} \sum_{j \neq i} w(\mathbf{z})_{i,j}^2) d\mathbf{z} =
$$
$$
\int_{\mathbf{z}} \sum_{i=1}^{D} (1 - w(\mathbf{z})_{i,i})^2 p(\mathbf{z}) + \gamma q(\mathbf{z}) w(\mathbf{z})_{i,i}^2 + \frac{1}{D} \sum_{j \neq i} (p(\mathbf{z}) + \gamma q(\mathbf{z})) w(\mathbf{z})_{i,j}^2 d\mathbf{z}
\tag{6}
$$

We can notice that the integrand is separable in each $w(\mathbf{z})_{i,j}$ thus resulting in an element-wise quadratic function for each $w(\mathbf{z})_{i,j}$. It is then easy to see that the optimal matrix $w(\mathbf{z})$ is diagonal with coefficients equal to:

$$
w(\mathbf{z})_{i,j} = \langle \phi_f(\mathbf{z})_i, \phi_{f'}(\mathbf{z})_j \rangle = \begin{cases} \frac{p(\mathbf{z})}{p(\mathbf{z}) + \gamma q(\mathbf{z})} & \text{if } i = j \\ 0 & \text{if } i \neq j \end{cases}
\tag{7}
$$

The solution equation 7 of the above relaxed problem is not necessarily reachable by any $f'$. We thus make the following assumption on the capacity of $f'$ :

**Assumption 1** *For any $\psi(\cdot)$, $f(\cdot)$, there exists $f'(\cdot)$ such that its feature maps $\phi_{f'}(\mathbf{z})_i$ satisfy the constraints in equation 7 for all $i, j \in \{1, \cdots, D\}$*

This assumption, akin to those made in margin-based methods like MDD which rely on expressive classifiers to estimate divergence, ensures theoretical alignment of source and target features. Importantly, as shown in Appendix A.6, our method remains empirically effective even when $f'$ has very limited capacity, suggesting that the assumption is not overly restrictive in practice.

**Lemma 1** *Under Assumption 1, the adversary $f'(\cdot)$ that satisfies equation 7 is a global minimum of the minimization problem of equation 5.*

Proof in Appendix A.1.1.

**Theorem 1** *Minimizers $\psi(\cdot)$ of the minimization problem of $\mathcal{L}_{task}$, equation 1, 2 align the marginal distributions of features in source and target domains i.e., $p(\mathbf{z}) = q(\mathbf{z})$ at optimum.*

Proof in Appendix A.1.2

Combining previous Lemma and Theorem provides the following result. At optimum, we have:

$$\forall z, \forall (i,j) \in \{1,...,D\} \times \{1,...,D\}, \quad \langle \phi_f(\mathbf{z})_i, \phi_{f'}(\mathbf{z})_j \rangle = \begin{cases} \frac{1}{1+\gamma} & \text{if } i = j \\ 0 & \text{if } i \neq j \end{cases} \qquad (8)$$

In Figure 2, we will show that this theoretical result is experimentally verified.

### 3.4 COMPARISON WITH MDD ZHANG ET AL. (2019)

Our method shares some architectural similarities with the MDD method proposed in Zhang et al. (2019). Both methods use a feature extractor and two heads, $f$ and $f'$. However, in Zhang et al. (2019), both networks $f$ and $f'$ have also a task-specific head and the method, originally conceived for classification only, aims at aligning the predicted probabilities of the task-specific heads using a cross-entropy loss. Contrary to that, the adversarial game we propose here concerns the correlation of the features extracted by $f$ and $f'$, which is thus independent on the task to perform. As a consequence, our method can be theoretically adapted to any architecture and task.

## 4 EXPERIMENTS

### 4.1 EXPERIMENTAL DETAILS

We perform three experiments to assess the performance of our method and compare it with several SOTA approaches. All experiments were conducted on an NVIDIA V100 GPU with 32GB memory. We demonstrate our approach on two tasks: classification and segmentation. A key advantage of our method over other SOTA approaches is that we make no assumptions about the type of domain shift. Accordingly, we evaluate it on different types of shifts: intensity variations (CT → MR, MNIST → MNIST-M, MNIST → USPS, USPS → MNIST) and geometric variations (CT → CBCT, DRIVE → CHASEDB1). Examples of these shifts across source and target datasets are provided in Appendix A.2. Further implementation details for reproducibility are given in Appendix A.8. Full details on all datasets used are provided in Appendix A.7.

**Digit classification.** We conduct UDA experiments on three digit datasets: MNIST (70,000 images), MNIST-M (149,002 images), and USPS (9,298 images). These datasets differ in color, texture, and background style, leading to domain shifts. We evaluate adaptation performance on MNIST → MNIST-M, USPS → MNIST, and MNIST → USPS.

**Retinal vasculature segmentation.** We evaluate on DRIVE (40 labeled images, source) and CHASEDB1 (28 labeled images, target). The datasets differ in field of view, optic disc localization, and vessel contrast.

**Liver segmentation.** We further assess our method on cross-modality liver segmentation. We consider two settings: CT → CBCT using a private dataset (15,827 CT slices, 13,024 CBCT slices) and CT → MR using LiTS (5,324 CT slices) as source and CHAOS (647 MR slices) as target. These settings capture both geometric (CT → CBCT) and intensity (CT → MR) shifts. All segmentation experiments are performed on 2D axial slices resampled to a common pixel size.

### 4.2 RESULTS

**Digit classification.** As showed in Table 1, we outperform all similar adversarial feature-alignment methods such as DANN Ganin et al. (2016) or ADDA Tzeng et al. (2017) by a large margin and contrastive approaches for feature-alignment such as Thota et al. (2021) or Avena et al. (2025). When the domain shift does not rely on geometric differences between source and target domains, as in MNIST → MNIST-M experiment, image alignment methods such as pixelDA Bousmalis et al. (2017) or DRANET Lee et al. (2021) can be effective. However, this hypothesis strongly restricts the use of such methods in unsupervised domain adaptation. Furthermore, they are often complex to implement as they require to train a generator, a discriminator, and a classifier. Our method achieves 98.6% on USPS → MNIST, outperforming the image-alignment DRANet baseline

Table 1: Performance evaluation of domain adaptation methods for digit classification using MNIST (M), MNIST-M (M-M), USPS (U) datasets. We indicate the best performance in **boldface**, while the second-best is underlined.

| Type | Method | Accuracy ↑ | | | Δ parameters ↓ |
|------|--------|------------|---|---|----------------|
| | | M → M-M | M → U | U → M | |
| | Source Only | 0.625 | 0.802 | 0.449 | 0% |
| **Feature Alignment** | DANN Ganin et al. (2016) | 0.851 | 0.851 | 0.730 | +4% |
| | MMD Long et al. (2015) | 0.769 | - | - | **+0 %** |
| | DSN Bousmalis et al. (2016) | 0.832 | 0.913 | - | +284% |
| | ADDA Tzeng et al. (2017) | 0.894 | 0.901 | 0.952 | +116% |
| | CDA Thota et al. (2021) | 0.602 | 0.942 | - | **+0%** |
| | SCoDA Avena et al. (2025) | 0.851 | 0.966 | - | **+0%** |
| | FARR (ours) | 0.982 | 0.980 | **0.986** | +42% |
| **Image Alignment** | CoGAN Liu & Tuzel (2016) | 0.620 | 0.912 | 0.891 | +1819% |
| | pixelDA Bousmalis et al. (2017) | 0.982 | 0.959 | - | +129% |
| | CyCADA Hoffman et al. (2018) | 0.921 | 0.956 | 0.965 | +1085% |
| | DRANET Lee et al. (2021) | **0.987** | **0.982** | 0.978 | +95% |
| | Target Only | 0.962 [a] | 0.978 | 0.991 | 0% |

[a]UDA methods can occasionally outperform target-only supervised training, as already reported in Lee et al. (2021)

(97.8%) by 0.8 percentage points. This notable gain appears on a UDA setting where the source domain has lower resolution and reduced style diversity, suggesting that our feature-level alignment method can be particularly effective under such conditions. The Δ parameters metric computes the proportion of parameters that is added for domain adaptation with respect to the original classification network. Image-alignment methods, such as CyCADA Zhu et al. (2017) or CoGAN Liu & Tuzel (2016), are composed of heavy blocks for domain adaptation. In our case, we only add **42%** of the parameters, as the adversary is only composed of the last layer block of the final layer group of the ResNet (as detailed in Appendix A.8), which is much less than all other image-alignment methods. In the context of digit classification, our method achieves **SOTA performance** across diverse domain shifts, while remaining **lightweight** compared to existing SOTA approaches. In Appendix A.10, we further demonstrate that our method can be adapted to transformer architectures in the context of digit classification.

In our ablation study of hyperparameters in Appendix A.4, we evaluate the robustness of our method with respect to the weighting parameters $\alpha$ and $\gamma$ on MNIST → MNIST-M. Although the best performance is achieved around $\alpha \approx 0.8$ and $\gamma \approx 0.05$ as described in Appendix A.8, we observe comparable results across a broad range, with $\alpha \in [0,1]$ and $\gamma \in [0,1]$. The $\alpha$ and $\gamma$ parameters mentioned were determined through hyperparameter search using Optuna Akiba et al. (2019). This indicates that the method is not overly sensitive to the precise choice of these hyperparameters.

**Retinal vasculature segmentation.** In Table 2, we show that our method outperforms all other SOTA methods both in AUC and F1 scores. Furthermore, we only need to add +2% parameters of the original U-Net segmentation model for the domain adaptation. Our method can thus effectively align features from source and target domains, even when they have important geometric and appearance differences. These findings confirm the **adaptability** of our method, demonstrating its successful extension to a **segmentation task** and its integration with a **distinct architecture**, namely U-Net.

**Liver segmentation.** Finally, in Table 3, we show that our method performs well also in the context of 2D liver segmentation with two different types of domain shift. It outperforms all other SOTA UDA methods, both based on feature alignment or self-training, by a wide margin. We also compare our method against a foundation model, SAM-Med 2D Cheng et al. (2023), in a zero-shot setting. SAM-Med 2D was trained on MR and CT images but not on CBCT data. This explains why it outperforms our method in CT → MR UDA but not on the CT → CBCT UDA problem, where its performance drops.

Table 2: Performance evaluation of domain adaptation methods for segmentation. Source: DRIVE. Target: CHASEDB1. We indicate the best performance in **boldface**, while the second-best is underlined.

| Type | Method | AUC ↑ | F1 ↑ | Δ parameters ↓ |
|---|---|---|---|---|
| **Self-Training** | SGL Zhou et al. (2021) | 0.978 | 0.718 | +100 % |
| | ProDA Zhang et al. (2021) | 0.971 | 0.699 | +102 % |
| | FR-UNet Liu et al. (2022) | 0.978 | 0.736 | **+0** % |
| | CDCL Chen et al. (2023) | 0.979 | 0.739 | - |
| | MILAL-PDSF Liu et al. (2024) | 0.981 | 0.740 | - |
| **Feature Alignment** | DisClusterDA Tang et al. (2022) | 0.977 | 0.737 | **+0** % |
| | FARR (ours) | **0.982** | **0.753** | +2 % |
| | Target Only | 0.989 | 0.842 | |

Table 3: Performance evaluation of domain adaptation methods for segmentation on two experiments. Source: CT, Target: CBCT or MR. We indicate the best performance in **boldface**, while the second-best is underlined.

| Type | Method | F1 score ↑ | |
|---|---|---|---|
| | | CT → CBCT | CT → MR |
| | Source Only | 0.541 | 0.548 |
| **Self-Training** | BDCL Lee et al. (2022) | 0.600 | - |
| **Feature Alignment** | DANN Fu et al. (2024) | 0.683 | 0.661 |
| | MDD Munk et al. (2024) | 0.700 | 0.681 |
| | FARR (ours) | **0.751** | 0.728 |
| **Zero-shot** | SAM-MED 2D Cheng et al. (2023) | 0.461 | **0.812** [a] |
| | Target Only | 0.903 | 0.868 |

[a]This model is trained fully supervised with MR labels

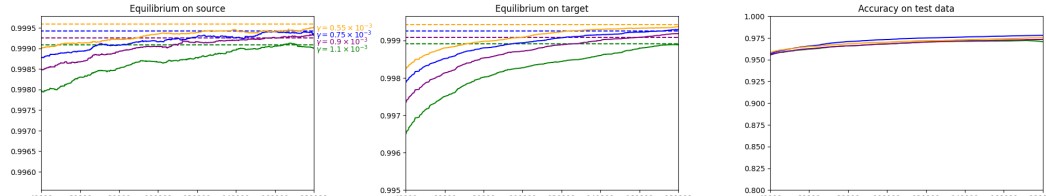

Figure 2: Convergence with respect to $\gamma$ parameter on MNIST $\rightarrow$ MNIST-M. The two plots on the left show the mean correlation values of $\phi_f(\mathbf{z})_i$ and $\phi_{f'}(\mathbf{z})_i$ ($y$ axis) over i with respect to the training iteration number ($x$ axis). The plot on the right shows the accuracy on test data from target domain ($y$ axis) with respect to the training iteration number ($x$ axis).

**Experimental confirmation of theoretical results.** As proved in equation 9, at convergence, the correlation of embeddings $\phi_f(\mathbf{z})_i$ and $\phi_{f'}(\mathbf{z})_j$, when $i = j$, should converge to $\frac{1}{1+\gamma}$. In Figure 2, we show that this theoretical result is verified experimentally. We trained 4 models using different values of $\gamma$ ($0.55 \times 10^{-3}, 0.75 \times 10^{-3}, 0.9 \times 10^{-3}, 1.1 \times 10^{-3}$) while fixing the other hyper-parameters ($\alpha$, batch size, learning rates) to the same values. As showed in Figure 2, we reach the equilibrium expected from equation 9 for both source and target domains. Indeed, equilibrium on source (resp. target) shows the average value of the correlation of $\phi_f(\mathbf{z})_i$ and $\phi_{f'}(\mathbf{z})_i$ over $i$, for features extracted from source (resp. target) images. These values are computed for each training iteration, with a batch size of 32. For each $\gamma$, these mean correlation values actually converge to $\frac{1}{1+\gamma}$. Further experiments about the cross-correlation matrix $\mathcal{C}[i,j]$ and representations spaces can be found in Appendix A.5 and Appendix A.3.

## 5    CONCLUSION

We proposed a novel feature-alignment approach for UDA based on redundancy reduction, and demonstrated its performance through qualitative and quantitative experiments on various datasets, with various types of domain shift. Our method is lightweight, and can be theoretically adapted to any architecture. It outperforms other state-of-the-art UDA approaches belonging to the same category on different datasets and tasks. Furthermore, the method is generic, makes no assumptions on the supervised task or on the type of domain shift, and it has reasonable theoretical guarantees validated experimentally.

**Limitations.** The results we provided on classification and segmentation tasks are appealing, but they were only obtained using 2D datasets. In medical image segmentation, using the entire 3D volumes usually improves the results. Nevertheless, we believe that our method could be easily adapted to 3D scans, using a 3D UNet for the segmentation tasks, for instance. Another limitation of this article is the lack of experiment on regression task. While our method is task-agnostic by design, current experiments are limited to classification and segmentation. Evaluating on regression tasks would further validate its generality.

**Future directions.** As our method makes no hypothesis about the task, we will investigate its performance in other tasks such as regression or object detection. Another research path might be the extension of this UDA method to the semi-supervised setting. Indeed, in some context, it is possible to benefit from a few number of labeled data in target domain. Eventually, we also plan to extend the proposed method to a Domain Generalization setting, leveraging multiple source domains.

**Reproducibility statement.** We ensure reproducibility by releasing the full codebase for training and evaluation, accompanied by detailed instructions [3]. All theoretical results and proofs are provided in the paper Section 3.3 and Appendix A.1. Comprehensive training procedures, hyperparameter configurations, and optimization details for all experiments are reported in Appendix A.8. The majority of datasets used are publicly available and described in Appendix A.7; two datasets are private but are documented in Appendix A.8.3 with all pre-processing and augmentation details. In addition, we demonstrate in Appendix A.4 that our method is robust to hyperparameter variations, which further facilitates reproducibility in practice.

---

[3] https://anonymous.4open.science/r/FARR-06B3

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

## A  APPENDIX

### A.1  PROOF OF THE THEORETICAL GUARANTEES

#### A.1.1  PROOF OF LEMMA 1

In this subsection, we provide the proof of the Lemma 1 of Section 3.3. First, let's recall equation 5 and equation 7 of the paper.

$$
\begin{aligned}
\arg\min_{f'} \mathbb{E}_{\mathbf{z}\sim p} &\left[ \sum_{i=1}^{D}(1 - \langle\phi_f(\mathbf{z})_i, \phi_{f'}(\mathbf{z})_i\rangle)^2 + \frac{1}{D}\sum_{i=1}^{D}\sum_{j\neq i}\langle\phi_f(\mathbf{z})_i, \phi_{f'}(\mathbf{z})_j\rangle^2 \right] \\
&+ \gamma\mathbb{E}_{\mathbf{z}\sim q}\left[ \sum_{i=1}^{D}\langle\phi_f(\mathbf{z})_i, \phi_{f'}(\mathbf{z})_i\rangle^2 + \frac{1}{D}\sum_{i=1}^{D}\sum_{j\neq i}\langle\phi_f(\mathbf{z})_i, \phi_{f'}(\mathbf{z})_j\rangle^2 \right]
\end{aligned}
\tag{5}
$$

$$
\langle\phi_f(\mathbf{z})_i, \phi_{f'}(\mathbf{z})_j\rangle = \begin{cases} \frac{p(\mathbf{z})}{p(\mathbf{z})+\gamma q(\mathbf{z})} & \text{if } i = j \\ 0 & \text{if } i \neq j \end{cases}
\tag{7}
$$

**Assumption 1** *For any $\psi(\cdot)$, $f(\cdot)$, there exists $f'(\cdot)$ such that its feature maps $\phi_{f'}(\mathbf{z})_i$ satisfy the constraints in equation 7 for all $i, j \in \{1, \cdots, D\}$*

**Lemma 1** *Under Assumption 1, the adversary $f'(\cdot)$ that satisfies equation 7 is a global minimum of the minimization problem of equation 5.*

Proof.

$$
\begin{aligned}
\forall f', \quad \mathbb{E}_{\mathbf{z}\sim p} &\left[ \sum_{i=1}^{D}(1 - \langle\phi_f(\mathbf{z})_i, \phi_{f'}(\mathbf{z})_i\rangle)^2 + \frac{1}{D}\sum_{i=1}^{D}\sum_{j\neq i}\langle\phi_f(\mathbf{z})_i, \phi_{f'}(\mathbf{z})_j\rangle^2 \right] \\
&+ \gamma\mathbb{E}_{\mathbf{z}\sim q}\left[ \sum_{i=1}^{D}\langle\phi_f(\mathbf{z})_i, \phi_{f'}(\mathbf{z})_i\rangle^2 + \frac{1}{D}\sum_{i=1}^{D}\sum_{j\neq i}\langle\phi_f(\mathbf{z})_i, \phi_{f'}(\mathbf{z})_j\rangle^2 \right] \\
&\geq \sum_{i=1}^{D}\int_{\mathbf{z}} p(z)(1 - \langle\phi_f(\mathbf{z})_i, \phi_{f'}(\mathbf{z})_i\rangle)^2 + \gamma q(z)\langle\phi_f(\mathbf{z})_i, \phi_{f'}(\mathbf{z})_i\rangle)^2 d\mathbf{z} \\
&= \sum_{i=1}^{D}\int_{\mathbf{z}} (p(\mathbf{z}) + \gamma q(\mathbf{z}))\langle\phi_f(\mathbf{z})_i, \phi_{f'}(\mathbf{z})_i\rangle)^2 - 2p(\mathbf{z})\langle\phi_f(\mathbf{z})_i, \phi_{f'}(\mathbf{z})_i\rangle + p(\mathbf{z})d\mathbf{z} \\
&\geq \sum_{i=1}^{D}\int_{\mathbf{z}} p(\mathbf{z}) - \frac{(-2p(\mathbf{z}))^2}{4(p(\mathbf{z}) + \gamma q(\mathbf{z}))}d\mathbf{z} \\
&= \sum_{i=1}^{D}\int_{\mathbf{z}} \frac{\gamma p(\mathbf{z})q(\mathbf{z})}{p(\mathbf{z}) + \gamma q(\mathbf{z})}d\mathbf{z}
\end{aligned}
\tag{9}
$$

Indeed, the minimum of a quadratic polynomial $aX^2 + bX + c$ with $a > 0$ is $c - \frac{b^2}{4a}$, and we actually have: $\forall\mathbf{z}, \forall\gamma > 0, p(\mathbf{z}) + \gamma q(\mathbf{z}) > 0$.

If this lower bound is reached by some $f'(\cdot)$, then it is a global minima of the minimization problem of equation 5.

Under Assumption 1, let's consider an adversary $f'(\cdot)$ that satisfies equation 7. We have:

$$\mathbb{E}_{\mathbf{z}\sim p}\left[\sum_{i=1}^{D}(1-\langle\phi_f(\mathbf{z})_i,\phi_{f'}(\mathbf{z})_i\rangle)^2+\frac{1}{D}\sum_{i=1}^{D}\sum_{j\neq i}\langle\phi_f(\mathbf{z})_i,\phi_{f'}(\mathbf{z})_j\rangle^2\right]$$

$$+\gamma\mathbb{E}_{\mathbf{z}\sim q}\left[\sum_{i=1}^{D}\langle\phi_f(\mathbf{z})_i,\phi_{f'}(\mathbf{z})_i\rangle^2+\frac{1}{D}\sum_{i=1}^{D}\sum_{j\neq i}\langle\phi_f(\mathbf{z})_i,\phi_{f'}(\mathbf{z})_j\rangle^2\right] \tag{10}$$

$$=\sum_{i=1}^{D}\int_{\mathbf{z}}p(\mathbf{z})(1-\frac{p(\mathbf{z})}{p(\mathbf{z})+\gamma q(\mathbf{z})})^2+\gamma q(\mathbf{z})(\frac{p(\mathbf{z})}{p(\mathbf{z})+\gamma q(\mathbf{z})})^2 d\mathbf{z}$$

$$=\sum_{i=1}^{D}\int_{\mathbf{z}}\frac{\gamma p(\mathbf{z})q(\mathbf{z})}{p(\mathbf{z})+\gamma q(\mathbf{z})}d\mathbf{z}$$

From equation 9 and equation 10, we can conclude that the adversary $f'(\cdot)$ that satisfies equation 7 is a global minimizer of the minimization problem of equation 5.

### A.1.2 PROOF OF THEOREM 1

In this subsection, we provide a proof of the Theorem 1 of Section 3.3. First, let's recall 1, 2.

$$\arg\min_{f'}\mathcal{L}_{\text{align}}\left(f(z^S),f'(z^S)\right)+\gamma\mathcal{L}_{\text{sep}}\left(f(z^T),f'(z^T)\right) \tag{1}$$

$$\arg\min_{\psi}\mathcal{L}_{\text{task}}(h(x^S),y^S)+\alpha\mathcal{L}_{\text{align}}\left(f(z^S),f'(z^S)\right)+\gamma\mathcal{L}_{\text{align}}\left(f(z^T),f'(z^T)\right) \tag{2}$$

**Theorem 1** *Minimizers $\psi(\cdot)$ of the minimization problem of $\mathcal{L}_{\text{task}}$, equation 1, 2 align the marginal distributions of features in source and target domains i.e., $p(\mathbf{z})=q(\mathbf{z})$ at optimum.*

Proof.

We suppose that the term $\mathcal{L}_{\text{task}}(h(x^S),y^S)$ is negligible. The task error on source domain is supposed to remain low during UDA with no restriction for the choice of $\psi$.

We are interested in the following minimization problem:

$$\arg\min_{\psi}\alpha\mathbb{E}_{\mathbf{z}\sim p}\left[\sum_{i=1}^{D}(1-\langle\phi_f(\mathbf{z})_i,\phi_{f'}(\mathbf{z})_i\rangle)^2+\frac{1}{D}\sum_{i=1}^{D}\sum_{j\neq i}\langle\phi_f(\mathbf{z})_i,\phi_{f'}(\mathbf{z})_j\rangle^2\right]$$

$$+\gamma\mathbb{E}_{\mathbf{z}\sim q}\left[\sum_{i=1}^{D}(1-\langle\phi_f(\mathbf{z})_i,\phi_{f'}(\mathbf{z})_i\rangle)^2+\frac{1}{D}\sum_{i=1}^{D}\sum_{j\neq i}\langle\phi_f(\mathbf{z})_i,\phi_{f'}(\mathbf{z})_j\rangle^2\right] \tag{11}$$

As $f'(\cdot)$ minimizes the optimization problem of equation 1, it satisfies equation 7. Adding the constraint $\int_z q(z)dz=1$ to ensure that $q$ is a density, the optimization problem becomes

$$\arg\min_{\psi}\sum_{i=1}^{D}\int_{\mathbf{z}}\left[(\alpha p(\mathbf{z})+\gamma q(\mathbf{z}))(1-\frac{p(\mathbf{z})}{p(\mathbf{z})+\gamma q(\mathbf{z})})^2 dz\right]-\lambda(\int_z q(z)dz-1) \tag{12}$$

At optimum, the following conditions must be verified:

$$\begin{cases}\gamma(1-\frac{p(\mathbf{z})}{p(\mathbf{z})+\gamma q(\mathbf{z})})^2+2(\alpha p(\mathbf{z})+\gamma q(\mathbf{z}))\frac{\gamma p(\mathbf{z})}{(p(\mathbf{z})+\gamma q(\mathbf{z}))^2}(1-\frac{p(\mathbf{z})}{p(\mathbf{z})+\gamma q(\mathbf{z})})-\lambda=0\\\int_{\mathbf{z}}q(\mathbf{z})d\mathbf{z}=1\end{cases}$$

$$\Leftrightarrow\begin{cases}-\lambda w^3+(2\alpha\gamma-3\lambda)w^2+3(\gamma-\lambda)w+(\gamma-\lambda)=0 \qquad \text{with } w=\frac{p(\mathbf{z})}{\gamma q(\mathbf{z})}\\\int_{\mathbf{z}}q(\mathbf{z})d\mathbf{z}=1\end{cases} \tag{13}$$

The polynomial of the first condition admits a solution. There exists $\lambda, w$ such that all the conditions are verified. In this case, we have two densities of probability $q$ and $p$ such that: $\forall z, p(\mathbf{z}) = \gamma w q(\mathbf{z})$. This implies $p(\mathbf{z}) = q(\mathbf{z})$. Therefore, there exists $\lambda$ such that the conditions of equation 13 are fulfilled, leading to $p(\mathbf{z}) = q(\mathbf{z})$.

This proves that the minimizers of $\mathcal{L}_{\text{task}}$, equation 1, 2 align the marginal distributions of features in source and target domains *i.e.*, $p(\mathbf{z}) = q(\mathbf{z})$ at optimum.

## A.2 TYPES OF DOMAIN SHIFTS


<div>
Source CT image

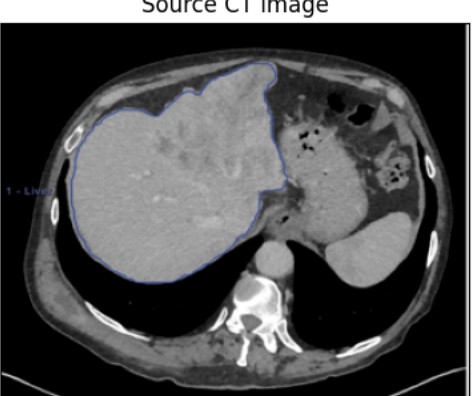
</div>
<div>
Target CBCT image

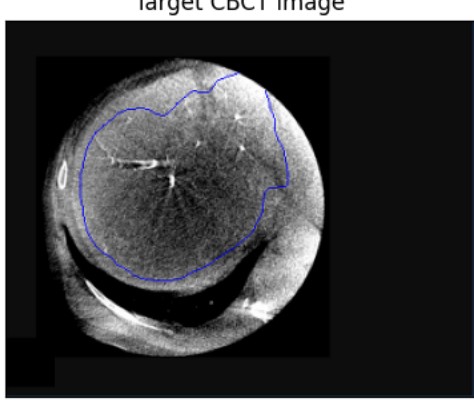
</div>
</div>

Figure 3: Left: Axial slice from abdominal CT (source domain). Right: Axial slice from abdominal CBCT (target domain) with intra-arterial injection of the same patient, with the same visualization window. Both come a from private dataset.

In Figure 3, we display CT (source domain) and CBCT (target domain) slices that were used in the UDA experiment for liver segmentation task. The images of this Figure are aligned for illustration purpose, but we used only unpaired slices in our UDA experiment. The blue border in each image demarcates the liver area. The corresponding quantitative results of this experiment are presented in the Table 3 of our paper. CBCT is different from the traditional CT as it entails a limited reconstructed field of view, specific artifacts and an intra-arterial injection of contrast medium. The domain shift between these two modalities is consequently composed of intensity and geometry differences.


<div>
Source CT image

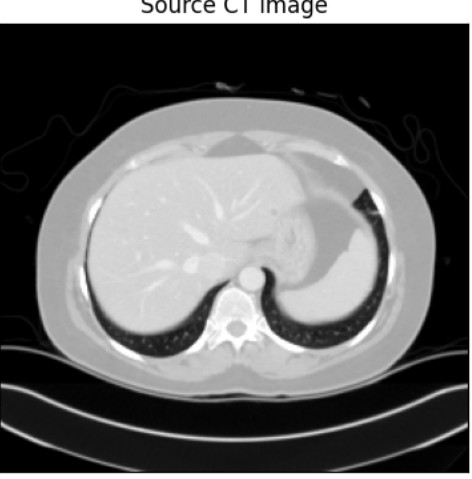
</div>
<div>
Target MR image

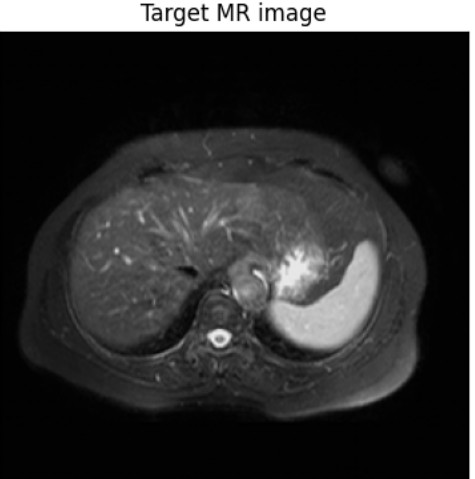
</div>
</div>

Figure 4: Left: Axial slice from abdominal CT from CHAOS + LiTS datasets (source domain). Right: Axial slice from abdominal MR from CHAOS dataset (target domain).

In Figure 4, we display CT (source domain) and MR (target domain) slices that were used in our UDA experiment for liver segmentation task. The corresponding quantitative results of this experiment are presented in the Table 3 of our paper. The domain shift between CT and MR images differs from the one observed between CT and CBCT, as previously discussed. While CT and MR modalities share comparable fields of view, they exhibit substantial differences in intensity distributions.

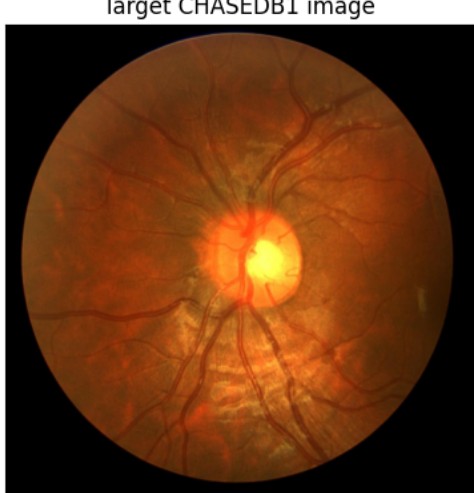

Figure 5: Left: Digital Retinal Image from DRIVE dataset (source domain). Right: Digital Retinal Image from CHASEDB1 dataset (target domain).

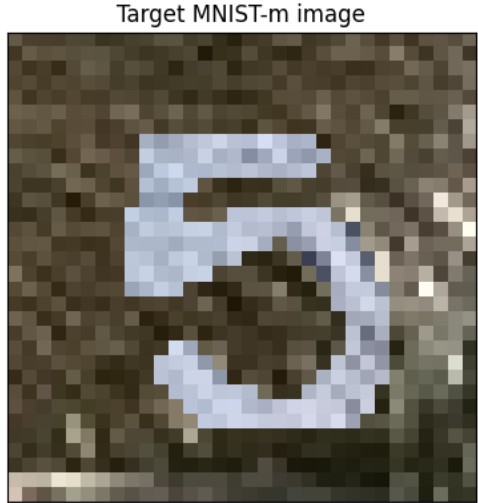

Figure 6: Left: Grayscale handwritten digits from MNIST dataset (source domain). Right: RGB handwritten digits from MNIST-M dataset (target domain).

In Figure 5, we display digital retinal images from DRIVE (source domain) and from CHASEDB1 (target domain) datasets that were used in our UDA experiment for retinal vasculature segmentation. The corresponding quantitative results of this experiment are presented in the Table 2 of our paper. As in CT to CBCT UDA, the domain shift is also composed with intensity and geometry differences. DRIVE images were captured with a wider field of view than CHASEDB1 images. Furthermore, the optic disk position is different between these source and target images. In DRIVE images, it appears on the left or on the right of the fundus region, whereas it is located on its center in CHASEDB1

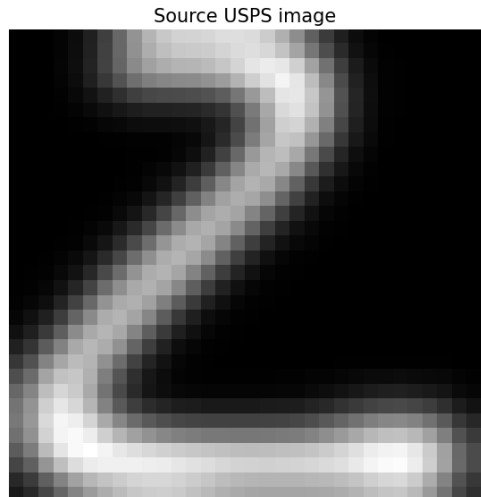 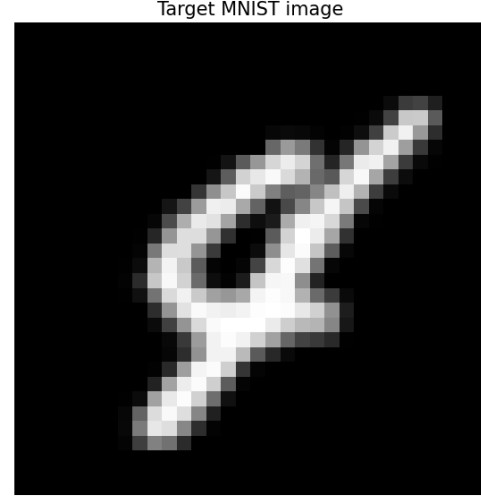

Figure 7: Left: Grayscale handwritten digits from USPS dataset (source domain). Right: Grayscale handwritten digits from MNIST dataset (target domain).

images. Finally, there is more intensity variations inside blood vessels regarding CHASEDB1 images comparing to DRIVE images.

In Figure 6, we display handwritten digits from MNIST (source domain) and MNIST-M (target domain) datasets that were used in our UDA experiment for digit classification. The corresponding quantitative results of this experiment are presented in the Table 1 of our paper. The MNIST and MNIST-M datasets exhibit notable differences in both color composition and intensity distribution. MNIST-M was derived by superimposing the original grayscale MNIST handwritten digits onto complex, natural image backgrounds sampled from RGB datasets. As a result, MNIST-M images are represented in three color channels (RGB), in contrast to the single-channel grayscale format of the original MNIST dataset. Furthermore, the backgrounds of MNIST-M images introduce visual noise and texture, making the classification task more challenging.

As shown in Figure 7, there is a clear domain shift between USPS and MNIST. USPS digits are collected at a lower resolution (16×16 compared to 28×28 in MNIST), tend to be written more neatly, and exhibit smoother, less varied handwriting styles. In contrast, MNIST digits are larger, noisier, and include a much wider diversity of handwriting.

In addition, USPS has far fewer images (around 9,000 samples compared to MNIST's 70,000), and the digit styles are less complex and less diverse. This means a model trained on USPS captures only a limited range of digit variations. When transferred to MNIST, the model encounters digit appearances it has never seen before (e.g., thicker strokes, unusual slants, or noisier backgrounds), leading to poor generalization.

Because of these differences, models trained on USPS (source) tend to struggle when applied to MNIST (target), as they fail to generalize to the broader variations present in MNIST. Conversely, the MNIST → USPS transfer direction is generally more successful, since MNIST's larger sample size and richer variability cover much of the USPS distribution. This asymmetry highlights the importance of both dataset size and distributional diversity in cross-domain generalization.

## A.3   FEATURE ALIGNMENT

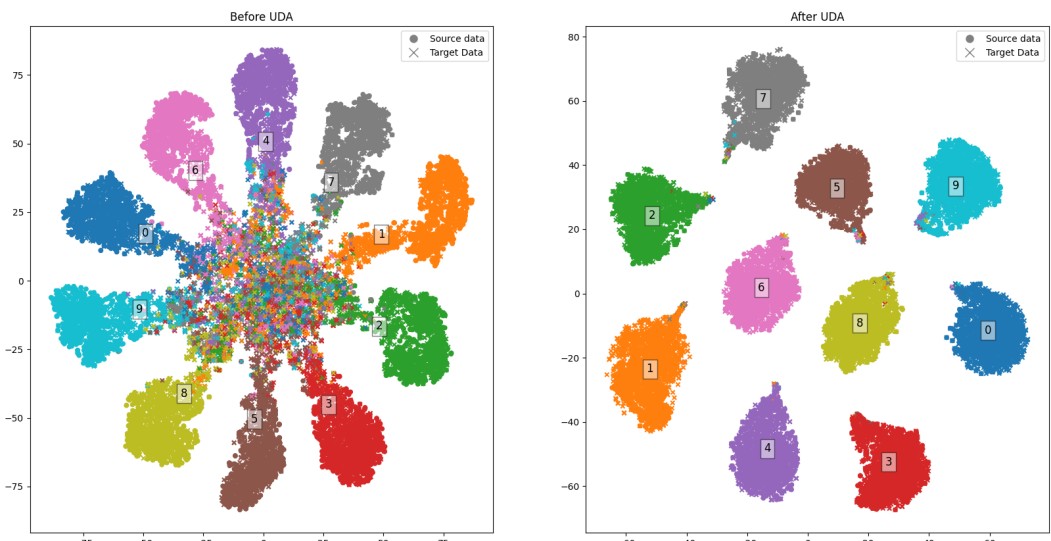

Figure 8: T-SNE showing source (MNIST) and target (MNIST-M) embeddings before (left) and after (right) applying our FARR UDA.

In Figure 8, we propose a qualitative evaluation of our FARR method on MNIST (source domain) to MNIST-M (target domain) UDA for classification task. We use t-distributed Stochastic Neighbor Embedding (t-SNE) to visualize the high-dimensional feature representations learned by the model. In this Figure, we can visualize the 2D projections of features extracted by $\psi$ from source and target images before (left) and after (right) UDA.

In Figure 8, the circles correspond to representations from source images and the crosses correspond to representations from target images. The class labels are represented by colors and by Figures in each plot.

This visualization reveals that the learned features are well-aligned between source and target data after applying our FARR UDA. Before UDA, the representations from source images are closed when they belong to the same class, and separated when they do not. However, the representations from target images are mixed between different classes. This qualitatively shows that the model trained on source domain only does not extract domain-invariant features. On the other hand, after UDA, samples from the target domain are mixed with their corresponding source counterparts. This suggests that the model has learned domain-invariant features. This alignment is critical for achieving good performance on the unlabeled target domain and indicates successful adaptation.

These results indicate that FARR successfully aligns features from source and target domains on the MNIST to MNIST-M UDA task.

## A.4   ABLATION STUDY ON THE SENSITIVITY OF OUR METHOD TO HYPERPARAMETERS $\alpha$, $\gamma$

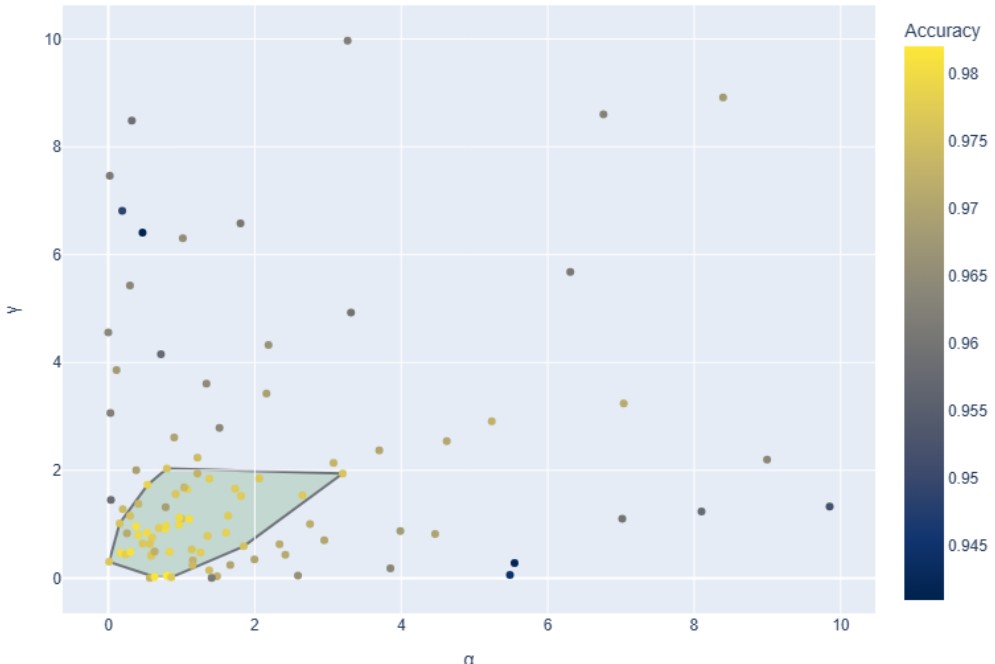

Figure 9: Ablation study on hyperparameters $\alpha$ and $\gamma$. Test accuracy on the MNIST-M target domain was evaluated following unsupervised domain adaptation from the MNIST source domain using the proposed method FARR. A heatmap was generated to illustrate the impact of varying the hyperparameters $\alpha$ (x-axis) and $\gamma$ (y-axis) on performance in the target domain. The results demonstrate that the method exhibits robustness to changes in these hyperparameters, with the highest accuracies observed when both $\alpha$ and $\gamma$ fall within the range [0, 1]. This region of stability is highlighted by a polygon that encloses the set of hyperparameter configurations yielding a target domain accuracy exceeding 0.975.

In Figure 9, we present a systematic study of the stability of our proposed method FARR with respect to the hyperparameters $\alpha$ and $\gamma$. To ensure comparability, we adopt the exact same experimental setup as described in the main paper, including both the model architecture and the optimization procedure. We then perform UDA training while varying $\alpha$ and $\beta$ over a broad interval, $[0, 10]$. The resulting performance on the target domain MNIST-M reveals two key observations. First, within the polygon-shaped region of the grid, the method consistently achieves accuracy above 0.975, indicating that small to moderate variations of $\alpha$ and $\gamma$ have negligible impact on performance. Second, even for extreme values at the boundaries of the search range, the lowest accuracy remains around 0.945. This is still higher than all feature-alignment baselines reported in the main article (Table 1). Taken together, these results demonstrate that the proposed method exhibits strong robustness to hyperparameter selection, making it practical and reliable in real-world scenarios where extensive tuning is often infeasible.

## A.5   REDUNDANCY REDUCTION

In this section, we provide experimental results related to the cross-correlation matrix $\mathcal{C}$ defined as
$\mathcal{C}[i, j] \triangleq \langle \phi_f(\mathbf{z})_i, \phi_{f'}(\mathbf{z})_j \rangle$.

| | Source Domain | Target Domain |
|---|---|---|
| Average on-diagonal | 0.9997 | 0.9998 |
| Average off-diagonal | 0.0025 | 0.0027 |

Table 4: Average on-diagonal and off-diagonal values of the cross-correlation matrix $\mathcal{C}$ obtained from 8.960 MNIST images (source domain) and 8.960 MNIST-M images (test domain) after UDA.

As presented in our paper Section 3.3, combining previous Lemma 1 and Theorem 1 provides the following result. At optimum, we have:

$$\forall z, \forall (i,j) \in \{1,...,D\} \times \{1,...,D\}, \quad \mathcal{C}[i,j] = \langle \phi_f(\mathbf{z})_i, \phi_{f'}(\mathbf{z})_j \rangle = \begin{cases} \frac{1}{1+\gamma} & \text{if } i = j \\ 0 & \text{if } i \neq j \end{cases} \quad (9)$$

In Table 4, we provide a table with average on-diagonal and off-diagonal values of the cross-correlation matrix $\mathcal{C}$, computed using 8.960 source images and 8.960 target images from the test datasets. The features were extracted using $\psi, f, f'$ after performing our FARR UDA training with hyperparameter $\gamma = 3.0 \times 10^{-4}$. From equation 9, we would expect to have an average on-diagonal value equal to 0.9997 and an average off-diagonal value close to 0 for both source and target domains. Our results in Table 4 confirm this theoretical result. In Figure. 10 and Figure 11, we provide examples of such cross-correlation matrices obtained for mnist $\rightarrow$ MNIST-M and USPS $\rightarrow$ MNIST UDA.

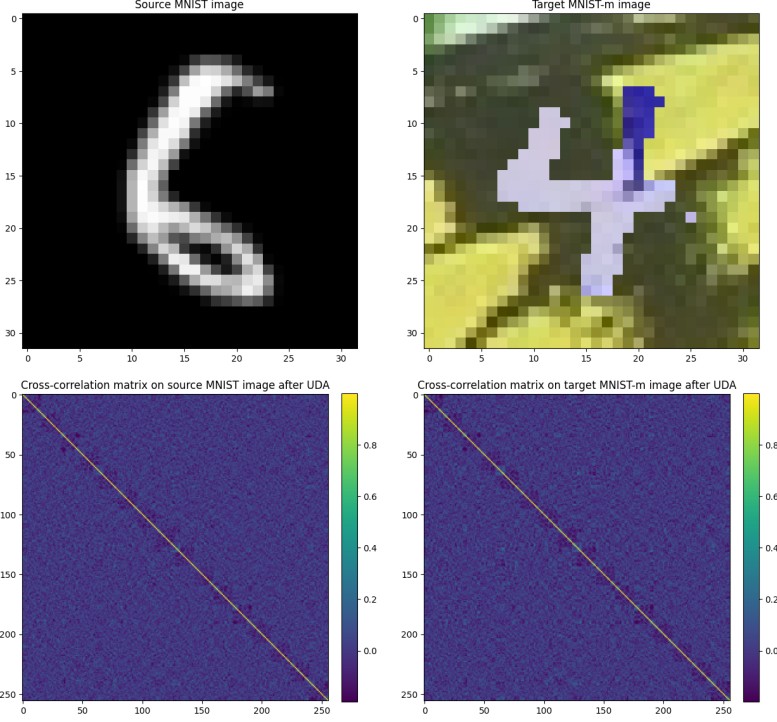

Figure 10: Example of MNIST (source domain) and MNIST-M (target domain) with the corresponding cross-correlation matrices after UDA.

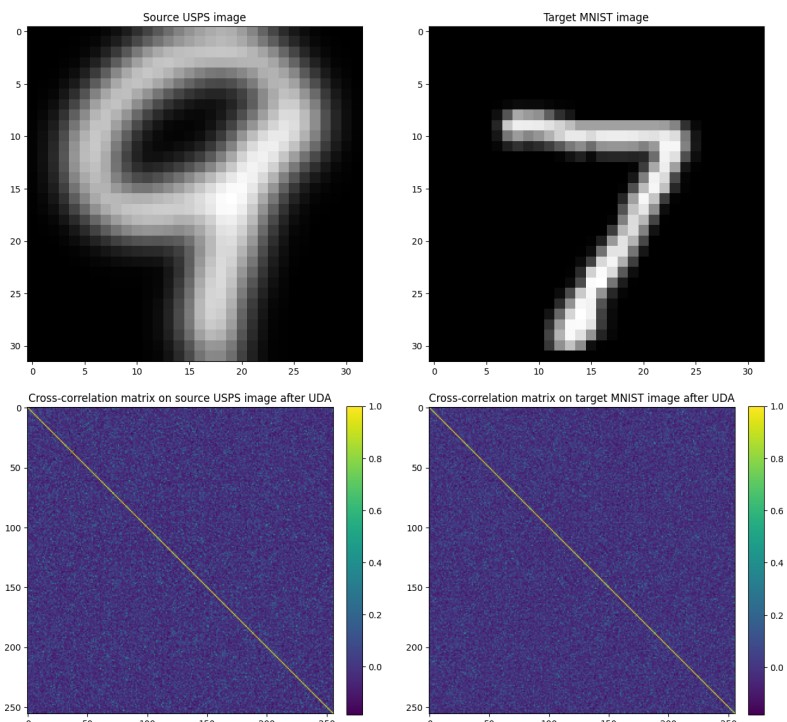

Figure 11: Example of USPS (source domain) and MNIST (target domain) with the corresponding cross-correlation matrices after UDA.

## A.6 EXPLORATION OF THE CAPACITY OF $f'$

In the main paper, we reported that the best classification performance regarding MNIST to USPS, USPS to MNIST, MNIST to MNIST-M was achieved when using the last block of a ResNet-18, which corresponds to approximately 42% of the overall network (Table 1). Since the theoretical guarantees of our method rely on the validity of Assumption 1—in particular, on the representational capacity of $f'$—we designed an additional experiment to explicitly examine the impact of the capacity of $f'$. The goal was to determine whether the method remains effective across a wide range of $f'$ sizes, including configurations with significantly reduced capacity compared to the one used in the main paper.

In this experiment, we defined $\psi$ as the full ResNet-18 backbone with an embedding size of 256. The function $f$ was implemented as a representation head composed of two linear layers with a ReLU activation in between (mapping from 256 to $f_{dim}$, and then from $f_{dim}$ to 32). The classifier $g$ was fixed as a linear layer from 32 to $n_{classes} = 10$. We pre-trained the model on MNIST source domain with $f_{dim} \in \{16, 32, 64, 128, 256\}$ and subsequently performed MNIST to MNIST-M UDA training by initializing $f'$ as a copy of $f$. Thus, the only factor that varies across experiments is $f_{dim}$, directly controlling the capacity of $f'$.

The results, summarized in Table 5, reveal two main trends. First, performance consistently improves as the capacity of $f'$ increases, highlighting the benefits of a richer representation. Second, and more importantly, even with minimal capacity—e.g., $f_{dim} = 16$, where $f'$ accounts for only 0.17% of the total network parameters—the method still achieves a strong accuracy of 97% on the target domain. For comparison, when $f'$ corresponds to 42% of the network (as in the main paper), the accuracy reaches 98.2%. These findings demonstrate that our method is highly stable with respect to the capacity of $f'$ and remains effective even under severe capacity constraints.

| $f_{dim}$ | 16 | 32 | 64 | 128 | 256 |
|---|---|---|---|---|---|
| **Δ parameters** | 0.17% | 0.33% | 0.66% | 1.33% | 2.66% |
| **Target Accuracy** | 0.9702 | 0.9708 | 0.9709 | 0.9716 | 0.9734 |

Table 5: Ablation study on $f'$ capacity. Test accuracy on the MNIST-M target domain was evaluated following unsupervised domain adaptation from the MNIST source domain using the proposed method FARR with various complexity of $f'$ taking $f_{dim} \in \{32, 64, 128, 256\}$ as described in section F. For each parameter $f_{dim}$, we report is the ratio between the number of parameters of $f'$ and the number of parameters of $\psi$ (**Δ parameters**) and the accuracy on the test set of the target domain (**Target Accuracy**).

## A.7 DATASETS

- **MNIST** LeCun et al. (1998): 28×28 grayscale images of handwritten digits (0–9), with 60,000 training and 10,000 test samples. Standard benchmark for digit classification.

- **MNIST-M** Ganin et al. (2016): Colored digits created by overlaying MNIST digits on random patches from BSDS500 images. Used for unsupervised domain adaptation to introduce style variability.

- **USPS** Denker et al. (1989): 16×16 grayscale images of handwritten digits, totaling 9,298 samples. Dataset for UDA experiments with limited style diversity.

- **DRIVE** Staal et al. (2004): 40 retinal images (20 training, 20 test) with manually annotated blood vessels. Each image is 565×584 pixels. Used for vessel segmentation tasks in medical imaging.

- **CHASEDB1** Hunter et al. (2013): 28 high-resolution retinal images (14 training, 14 test) with annotated vessel masks. Images are 999×960 pixels. Another benchmark for retinal vessel segmentation.

- **CHAOS** Kavur et al. (2021): Combined CT and MR images of healthy abdominal organs. Dataset includes multiple organs (liver, spleen, kidneys) with manual segmentations. Used for multi-organ segmentation tasks.

- **LiTS** Bilic et al. (2019): Liver Tumor Segmentation dataset containing 131 CT scans with liver and tumor annotations. Standard benchmark for liver and tumor segmentation in abdominal CT imaging.

## A.8 TRAINING DETAILS

We provide the detailed training protocols to ensure reproducibility. Unless otherwise stated, all experiments were conducted on a single Nvidia V100 GPU with 32GB memory. For each dataset pair, we adopted a four-fold split: $\frac{3}{4}$ of the data for training and validation, and $\frac{1}{4}$ for testing. All models were trained with the Adam optimizer. For all experiments, the hyperparameters $\alpha$ and $\gamma$ were determined through hyperparameter search using Optuna Akiba et al. (2019).

### A.8.1 DIGIT CLASSIFICATION: MNIST→MNIST-M, MNIST→USPS, MNIST→USPS

- **Backbone:** ResNet18 with 32 channels.
- **Architecture:** $f'$ defined as the last layer block of the final layer group of ResNet-18, $g$ as the classification head. An ablation study on other decomposition can be found in Appendix A.9.
- **Augmentation:** None.
- **Hyperparameters:** Learning rate 1e−3, batch size 32. $\alpha$ and $\gamma$ can be found in Table 6.

### A.8.2 RETINAL VASCULATURE SEGMENTATION: DRIVE→CHASEDB1

- **Backbone:** 5-stage U-Net, 64 channels at stage 1.
- **Architecture:** $f'$ corresponds to the last stage of the U-Net, $g$ is the final $1 \times 1$ convolution.
- **Preprocessing:** Images resized to $512 \times 512$ pixels.

Table 6: Hyperparameters $\alpha$ and $\gamma$ for each UDA task.

| Task | $\alpha$ | $\gamma$ |
|------|------|------|
| MNIST $\to$ MNIST-M | 0.80 | $5.0 \times 10^{-2}$ |
| USPS $\to$ MNIST | 0.55 | $2.1 \times 10^{-2}$ |
| MNIST $\to$ USPS | 0.60 | 1.32 |

- **Augmentation:** Rotation, zoom, cut-mix.

- **Hyperparameters:** Learning rate 1e−3, $\alpha = 1.5 \times 10^{-1}$, $\gamma = 3.2 \times 10^{-3}$, batch size 4.

### A.8.3 LIVER SEGMENTATION

We used the same U-Net backbone as in the retinal experiments. All 2D axial slices were resampled to a common pixel size of 1.8mm. Augmentations included rotation, zoom, random flipping, contrast adjustment, and additive noise.

**CT→CBCT (Private dataset)**

- **Dataset:** 15,827 CT slices and 13,024 CBCT slices.

- **Hyperparameters:** Learning rate 1e−3, $\alpha = 2.4 \times 10^{-1}$, $\gamma = 6.2 \times 10^{-3}$, batch size 8.

**CT→MR (LiTS → CHAOS)**

- **Dataset:** 5,324 CT slices (LiTS) and 647 MR slices (CHAOS).

- **Hyperparameters:** Learning rate 1e−3, $\alpha = 3.0 \times 10^{-1}$, $\gamma = 7.5 \times 10^{-3}$, batch size 8.

### A.9 ABLATION STUDY ON NETWORK DECOMPOSITION FOR FARR

To better understand the flexibility of our approach, we conduct an ablation study on how the decomposition of the predictor $h(x) \equiv g(f(\psi(x)))$ into a feature extractor $\psi(\cdot)$, a representation head $f(\cdot)$, and a task-specific prediction head $g(\cdot)$ affects performance. Using ResNet-18 for the classification task, we vary the granularity of the representation head $f(\cdot)$. In particular, we compare using the entire `layer4` block (`self.layer4`) versus only its final sub-block (`self.layer4[1]`) as $f(\cdot)$, while adjusting the feature extractor $\psi(\cdot)$ accordingly. The size of $f$ and $f'$ consequently ranges from approximately 75% to 42% of the total number of parameters in the network. We conduct this experiment on USPS $\to$ MNIST using same hyperparameters ($\alpha$, $\gamma$, learning rate, batch size) for both decompositions.

In Table 7, this experiment reveals that even shallow representation heads can yield strong alignment. However, the relative capacity of $\psi$ and $f$ also determines the strength of the adversarial signal provided by $f'$, and thus influences the overall feature alignment. This highlights the importance of balancing the expressiveness of the feature extractor and the adversary, and confirms the robustness of our method to architectural choices.

| size of $\psi$ | 24.85% | 57.7% |
|------|------|------|
| size of $f$ | 75.05% | 42.2% |
| **Target Accuracy** | 0.961 | 0.986 |

Table 7: Ablation study on network decomposition for FARR in USPS $\to$ MNIST UDA for classification. The size of $\psi$ and $f'$ represent the number of parameters of the two modules divided by the total number of parameters of the network. Test accuracy on the MNIST target domain was evaluated using the proposed method FARR with various decompositions for $\psi$, $f$, $g$. As $g$ is fixed and represent 0.1% of the network size in both cases, we only report the size of $\psi$ and $f$.

Table 8: Test accuracy on the MNIST target domain was evaluated following unsupervised domain adaptation from the USPS source domain using the proposed method FARR with various backbones.

| **Backbone** | ResNet-18 | ViT-S/14 |
|---|---|---|
| **Target Accuracy** | 0.986 | 0.953[4] |

## A.10   FLEXIBILITY OF ARCHITECTURE

In our main experiments, we adopt CNN backbones: ResNet-18 for digit classification and U-Net for segmentation tasks (e.g., DRIVE→CHASEDB1, CT→CBCT, CT→MR). To further demonstrate the architectural flexibility of FARR, we conducted an experiment using a transformer backbone by replacing ResNet-18 with DINOv2-Small Oquab et al. (2023) (ViT-S/14) on the USPS→MNIST classification task (Table 8). FARR achieved 95.3% accuracy with DINOv2-Small, compared to 98.6% with ResNet-18. DINOv2 was primarily used to leverage pretraining and provide a strong initialization for the source-only model, analogous to using ResNet-18 with a low learning rate. FARR was implemented with the same hyperparameters ($\alpha$ and $\gamma$) as for ResNet-18 (Appendix A.8), and the representation head $f$ and adversary $f'$ were defined as the last transformer block of ViT-S/14. While this configuration—mirroring the ResNet-18 setup—may not be optimal for the transformer, these results indicate that FARR can be applied beyond CNNs, supporting its architecture-agnostic nature.

---

[4]For a fair comparison with ResNet-18, a full hyperparameter search ($\alpha$, $\gamma$, $\psi$ and $f$ split) should be performed as for ResNet-18

