# OpenReview forum: "Unsupervised domain adaptation via feature alignment and redundancy reduction"
_ICLR.cc/2026/Conference — ICLR 2026 Conference Withdrawn Submission_

### Official Review · Reviewer_z8nd · 2025-10-25

**Soundness:** 2
**Presentation:** 2
**Contribution:** 2
**Rating:** 2
**Confidence:** 4

**Summary:**

The authors propose FARR (Feature Alignment and Redundancy Reduction), a task-agnostic and lightweight feature-alignment method to address UDA problems.

**Strengths:**

- the paper is generally easy to follow
- the experiments include not only classification but segmentation tasks

**Weaknesses:**

__Major Concerns:__
- limited novelty:
  - the idea is basically the same to MCD [1], which minimizes the disagreement between $f,f'$ on $S$ (corresponding to $L_{align}$) and maximizes it on $T$ (corresponding to $L_{sep}$) w.r.t. $f'$, then minimizes the entire loss w.r.t. $\phi$ adversarially; the only difference is MCD uses $L_1$ to measure the disagreement while this work uses cosine distance.
   - however, this work fails to explains the advantage of cosine distance.
- insuficient justification:
   - this woks claims pervious works strongly rely on the hypothesis of domain shift types, which is not true.
   - most of previous works are designed based on [2], which indeed assumes a small joint error that may fail in large domain shift tasks; however, there is no additional assumption on domain shift types.
   - in contrast, the experiments conducted in this work are quite limited to segmentation, while popular DA benchmarks (Office-Home, DomainNet, ViSDA) are overlooked which contains various types of domain shifts.
   - in fact, compared to previous works, there is not much difference regarding the algorithm that can be decomposed as minimized source error and marginal distribution alignment, which is suggested since [2].
   - the author claims the method is task-agnostic; however, the performance on regression tasks are not tested.
   - besides, this works does not provide any analysis regarding the generalization error.
- marginal improvement:
   - for instance, in Tab.2, the improvement is only 0.001; in Tab.3, the proposed method is outperformed by zero-shot method.
   - besides, the comparisons such as MDD and DANN are a little bit old.


__Minor Concerns:__
- too much space left in page 8, which could be filled with more discussions or experiments.
- it would be better to include results of SAM in Tab.2.
- the UDA task itself becomes less popular due to emergence of foundation models since SFDA methods such as [3] can already outperform UDA methods.

***
[1] Maximum Classifier Discrepancy for Unsupervised Domain Adaptation, CVPR 2018

[2] A theory of learning from different domains,  Machine Learning 2010

[3] Source-Free Domain Adaptation with Frozen Multimodal Foundation Model, CVPR 2024

**Questions:**

see above

---

### Official Review · Reviewer_hSLD · 2025-10-30

**Soundness:** 2
**Presentation:** 1
**Contribution:** 2
**Rating:** 2
**Confidence:** 3

**Summary:**

This paper introduces FARR (Feature Alignment and Redundancy Reduction), a novel unsupervised domain adaptation (UDA) method designed to align source and target features through redundancy reduction. Unlike prior feature- or image-alignment approaches that assume specific types of domain shifts or require complex architectures (e.g., GANs), FARR is task-agnostic, lightweight, and theoretically grounded.

The key idea is to decompose the model into a feature extractor, a representation head, and a prediction head, and then introduce an adversarial head trained with redundancy-reduction losses (inspired by Barlow Twins) to encourage invariant features across domains. The paper provides theoretical proofs guaranteeing marginal feature alignment under mild assumptions and demonstrates effectiveness across diverse UDA tasks—digit classification, retinal segmentation, and cross-modality liver segmentation—showing consistent improvements over adversarial and self-training baselines.

**Strengths:**

+ The resulting framework is simple yet general, avoiding task- or architecture-specific losses.
+ Theoretical grounding: this paper provides theoretical support.
+ FARR achieves superior or comparable performance to strong baselines (DANN, MDD, DRANet, SCoDA, etc.) while being parameter-efficient.

**Weaknesses:**

1) The paper lacks direct comparison with recent transformer-based or contrastive UDA approaches beyond Avena et al. (2025). Including more contemporary baselines (e.g., CLIP-based or DINOv2-based adaptation) could clarify competitiveness in modern architectures.

2) Visualization of aligned feature distributions (e.g., t-SNE) could help illustrate the benefit of redundancy reduction intuitively.

3) The paper is not well organized. In Introduction, there is lack of introduction of the proposed method. In experiment section, the tables are not well organized.

4) What is the advantage of this method compared with alignment based methods (related work section)?

5) The paper is somewhat over-claimed. Traditional methods such as DANN, MMD are also agnostic to the type of domain shift, and could task-adaptable. However, the authors claim there is no method that ....

6) The method is not clearly presented. (1) and (2) only introduced how the loss is designed but did not explain why such design is reasonable.

7) Section 3.1 could be introduced ahead (1) and (2).

8) In (3), if only use the first term, how about the performance? In (4), if only use the first term, how about the performance?

**Questions:**

See weakness, especially 6).

---

### Official Review · Reviewer_JF6r · 2025-10-31

**Soundness:** 2
**Presentation:** 3
**Contribution:** 1
**Rating:** 2
**Confidence:** 3

**Summary:**

The paper proposes FARR, a unsupervised domain adaptation (UDA) method that leverages feature-alignment strategy based on redundancy reduction, that is task-adaptable and agnostic to the type of domain shift. The paper provides theoretical analysis of the proposed method and also conducts empirical analysis on classification and segmentation tasks.

**Strengths:**

1. The task-adaptable nature of the proposed method is well motivated and is important for real-world use cases.

2. The proposed methodology is well-explained and the overall paper is well written.

**Weaknesses:**

1. Limited novelty: As discussed in Sec. 3.4, the dual-classifier strategy closely resembles MDD (Zhang et al., 2019), and the redundancy-reduction objective in Sec. 3.1 is similar to the Barlow Twins mechanism (Zbontar et al., 2021), which is well studied. Please clarify what is fundamentally new in the proposed formulation and why these differences matter empirically or theoretically.

2. Theoretical contribution is unclear: Lines 219–220 claim guarantees for aligning the marginal distribution of features. Does this imply the guarantees do not extend to conditional or joint domain shifts? If they do, please state the assumptions.

3. Limited evaluation for classification: The proposed method is evaluated only on the simple Digits datasets for classification task. Datasets like OfficeHome [R1], VisDA [R2], DomainNet [R3] are standard benchmarks for classification and results on atleast one of these is required.

4. Missing baselines: Given the similarity to MDD, a comparison, at least for classification, with MDD would be important. Similarly, other recent UDA baselines are missing from the comparison [R4-R5].

5. Computational complexity analysis missing: A detailed training and inference computational complexity analysis is required to understand the practicality of the proposed method. Please also compare the complexity with relevant baselines.

6. The paper states that the method is agnostic to the type of domain shift, but no supporting analysis is provided. Since shifts can be marginal, conditional, or joint, an evaluation, at least on synthetic datasets, would help demonstrate performance across shift types.

[R1] Venkateswara, Hemanth, et al. "Deep hashing network for unsupervised domain adaptation." Proceedings of the IEEE conference on computer vision and pattern recognition. 2017.

[R2] Peng, Xingchao, et al. "Visda: The visual domain adaptation challenge." arXiv preprint arXiv:1710.06924 (2017).

[R3] Peng, Xingchao, et al. "Moment matching for multi-source domain adaptation." Proceedings of the IEEE/CVF international conference on computer vision. 2019.

[R4] Rangwani, Harsh, et al. "A closer look at smoothness in domain adversarial training." International conference on machine learning. PMLR, 2022.

[R5] Zhang, Xinyu, Meng Kang, and Shuai Lü. "Low category uncertainty and high training potential instance learning for unsupervised domain adaptation." Proceedings of the AAAI conference on artificial intelligence. Vol. 38. No. 15. 2024.

**Questions:**

1. Please clarify what is fundamentally new in the proposed formulation compared to Barlow Twins mechanism (Zbontar et al., 2021).

2. Lines 219–220 claim guarantees for aligning the marginal distribution of features. Does this imply the guarantees do not extend to conditional or joint domain shifts? If they do, please state the assumptions.

3. Report results of the proposed method on atleast one of the benchmark image classification datasets (OfficeHome, VisDA, DomainNet) and also compare with relevant baselines (MDD, [R4], [R5]).

4. A detailed training and inference computational complexity analysis is required to understand the practicality of the proposed method.

5. Since domain shifts can be marginal, conditional, or joint, an evaluation, at least on synthetic datasets, would help demonstrate performance across shift types.

---

### Official Review · Reviewer_Dp7Y · 2025-10-31

**Soundness:** 2
**Presentation:** 2
**Contribution:** 1
**Rating:** 2
**Confidence:** 4

**Summary:**

This paper proposes FARR, a novel method for unsupervised domain adaptation (UDA) that introduces a feature-alignment strategy based on redundancy reduction. The method is task-agnostic and adaptable to various domain shifts, supported by both theoretical guarantees and empirical evaluations. FARR outperforms existing feature-alignment approaches and remains competitive with state-of-the-art UDA methods on multiple datasets, demonstrating its effectiveness across classification and segmentation tasks.

**Strengths:**

1.The paper provides a solid theoretical guarantee for the feature alignment process, which is crucial for justifying the method's reliability.
2.The method is adaptable to different tasks (e.g., classification and segmentation) and does not make strong assumptions about the domain shift.
3.The paper provides detailed implementation details, including the release of code and configurations, which enhances reproducibility and encourages future research in this area.

**Weaknesses:**

1.The proposed FARR method, though task-agnostic, rehashes concepts that have already been explored in existing methods. For example, MDD shares similar goals of aligning feature representations between source and target domains, but with a margin-based loss. While the paper offers theoretical guarantees for FARR, this novelty does not translate into a significant departure from existing adversarial methods.
2.The experimental validation in the paper is largely confined to the MNIST dataset, which limits the generalizability of the proposed method. While MNIST is a widely used benchmark for domain adaptation, it is considered a relatively simple dataset with minimal domain shift compared to more challenging and complex datasets such as Office-Home, Office31, VisDA-2017, and DomainNet.   Without testing on these datasets, the claims of the method’s effectiveness across various domain shifts remain unsubstantiated. The authors should conduct additional experiments on these more complex benchmarks to better demonstrate the versatility and strength of their approach in handling more difficult domain adaptation tasks.
3.There are several formatting issues within the paper, such as the placement of table (e.g., Table 1 extending beyond the page width) and excessive whitespace between Table 2 and Table 3. These distract from the otherwise well-organized content and may hinder readability.

**Questions:**

1.Given that the paper primarily focuses on feature alignment, how do the results change when compared with other SOTA feature alignment methods that use contrastive learning or more complex adversarial strategies?
2.Could the observed improvements be attributed to stronger priors from the use of redundancy reduction (via Barlow Twins) rather than genuine domain alignment? A more detailed analysis could help clarify the true contribution of this component.
3.Why was MDD not included as a baseline method in the experiments? Does the FARR method offer improvements over MDD in the types of domain shifts explored in the paper?

---

### Note · Authors · 2025-11-12

I have read and agree with the venue's withdrawal policy on behalf of myself and my co-authors.